# The Effects of Peptide Receptor Radionuclide Therapy on the Neoplastic and Normal Pituitary

**DOI:** 10.3390/cancers15102710

**Published:** 2023-05-11

**Authors:** Pedro Marques

**Affiliations:** 1Pituitary Tumor Unit, Endocrinology Department, Hospital CUF Descobertas, 1998-018 Lisbon, Portugal; pedro.miguel.sousa.marques@gmail.com; 2Faculdade de Medicina, Universidade Católica Portuguesa, 2635-631 Lisbon, Portugal

**Keywords:** pituitary neuroendocrine tumour (PitNET), pituitary adenoma, peptide receptor radionuclide therapy (PRRT), somatostatin receptors (SSTRs), somatostatin receptor ligand (SRL)

## Abstract

**Simple Summary:**

Few therapies are available for patients with aggressive or metastatic pituitary neuroendocrine tumours (PitNETs). Peptide receptor radionuclide therapy (PRRT), widely used to treat gastroenteropancreatic neuroendocrine tumours, has been emerging as a potential option for aggressive or metastatic PitNETs if other treatment approaches are not feasible or failed in controlling the disease progression. However, the data regarding the use of PRRT in this setting is scarce, and mainly derives from few case reports or small series of cases. Here, we review the published data regarding the effectiveness and safety of PRRT in the management of aggressive or metastatic PitNETs, as well as the effects of PRRT on the pituitary function in other cancer patients.

**Abstract:**

Pituitary neuroendocrine tumours (PitNETs) are usually benign and slow-growing; however, in some cases, they may behave aggressively and become resistant to conventional treatments. Therapeutic options for aggressive or metastatic PitNETs are limited, and currently mainly consist of temozolomide, with little experience of other emerging approaches, including peptide receptor radionuclide therapy (PRRT). Somatostatin receptor expression in PitNETs explains the effectiveness of somatostatin analogues for treating PitNETs, particularly those hypersecreting pituitary hormones, such as growth hormone or adrenocorticotropic hormone. The expression of such receptors in pituitary tumour cells has provided the rationale for using PRRT to treat patients with aggressive or metastatic PitNETs. However, the PRRT efficacy in this setting remains unestablished, as knowledge on this today is based only on few case reports and small series of cases, which are reviewed here. A total of 30 PRRT-treated patients have been thus far reported: 23 aggressive PitNETs, 5 carcinomas, and 2 of malignancy status unspecified. Of the 27 published cases with information regarding the response to PRRT, 5 (18%) showed a partial response, 8 (30%) had stable disease, and 14 (52%) had progressive disease. No major adverse effects have been reported, and there is also no increased risk of clinically relevant hypopituitarism in patients with pituitary or non-pituitary neuroendocrine tumours following PRRT. PRRT may be regarded as a safe option for patients with aggressive or metastatic PitNETs if other treatment approaches are not feasible or have failed in controlling the disease progression, with tumour shrinkage occurring in up to a fifth of cases, while about a third of aggressive pituitary tumours may achieve stable disease. Here, the data on PRRT in the management of patients with aggressive pituitary tumours are reviewed, as well as the effects of PRRT on the pituitary function in other PRRT-treated cancer patients.

## 1. Introduction

Pituitary neuroendocrine tumours (PitNETs) originate from the adenohypophysis cells, and account for about 15% of all intracranial neoplasms [1,2]. Although PitNETs are usually benign, up to 30–45% of them invade the cavernous or sphenoid sinus [1,3,4], and occasionally, they also behave aggressively, invading other surrounding tissues, growing rapidly, recurring multiple times and/or becoming resistant to conventional treatments. Rarely, PitNETs may also metastasise [1,2].

Nuclear medicine employs radiopharmaceuticals for imaging and treatment purposes, and plays a key role in the management of patients with endocrine-related cancers, most notably neuroendocrine tumours (NETs). The therapeutic approach based on peptide receptor radionuclide therapy (PRRT), specifically directed to neoplasms expressing high levels of somatostatin receptors (SSTRs), has proven effective and safe in patients with NETs, and has now been emerging as a potential treatment option also for patients with aggressive PitNETs or pituitary carcinomas, now known as metastatic PitNETs.

Here, the existing clinical data regarding the effectiveness and safety of targeted PRRT in the management of aggressive or metastatic PitNETs are reviewed, as well as the effects of PRRT on the pituitary function in other cancer patients submitted to PRRT.

## 2. Principles, Usefulness and Safety of Peptide Receptor Radionuclide Therapy (PRRT)

PRRT is designed to deliver cytotoxic radiation locally and selectively, comprising a radionuclide linked by a chelator to a somatostatin receptor ligand that binds cell surface SSTRs (Figure 1). The two most used radiopeptides for therapeutic purpose are ^90^Y-DOTATOC (90Yttrium-DOTA-tyr3-octreotide) and ^177^Lu-DOTATATE (177Lutetium-DOTA-tyr3-octreotate) [5,6]. On the other hand, functional SSTR imaging using gamma-emitting isotopes such as ^68^Ga or ^111^In may be of great theranostic interest, as it may predicts the behaviour and effectiveness of therapeutic radiopharmaceuticals, and also allow biodistribution assessments, beyond its recognised usefulness in the diagnosis and staging of several cancers, particularly NETs [6,7].

DOTANOC shows a very high affinity for SSTR2 and low affinity for SSTR3 and SSTR5, whereas DOTATATE only binds to SSTR2; thus, the main target of PRRT with radiolabelled somatostatin analogues is represented by SSTR2 [5,7]. PRRT has been successfully used to treat NETs due to their high expression levels of SSTR2 [8,9,10], but PRRT may also be effective for other cancers, such as primary brain tumours [5], paragangliomas [11,12,13,14], or thyroid cancer [15,16,17,18].

PRRT has emerged as a crucial treatment for advanced NETs [8,9,10,19], and it is now approved for metastatic SSTR-positive gastroenteropancreatic NETs based on the NETTER-1 trial [10]. PRRT is effective and safe when employed in the re-treatment of NET patients [20], and also for bronchopulmonary NETs expressing SSTRs [21]. The high expression of SSTRs in other tumours has expanded the clinical use of PRRT [19,22]. Brain tumours may express high levels of SSTRs, in particular meningiomas (90% of them express SSTR2) [5,23,24,25], but also astrocytomas and gliomas [5,26,27,28,29], making such tumours amenable to PRRT. PRRT has also been effective for treating paragangliomas, with disease control rates of 67–80% [12,13], including a malignant primary sellar paraganglioma case [11]. Additionally, PRRT has been used with variable efficacy for medullary and non-medullary thyroid cancer [15,16,17,18]. 

Because PRRT is a targeted therapy, the risk of systemic adverse effects is lower compared to other conventional forms of radiotherapy or chemotherapy. In general, PRRT is safe; however, there are few side effects that may occur. Nephrotoxicity is the most common adverse effect due to the kidney excretion of the radiolabelled somatostatin analogues. Amin-acid infusion should be given in order to reduce the renal uptake of radiopharmaceuticals, and thereby to minimise the occurrence of kidney injury [5,19,22]. Other common side effects include transitory and often mild haematotoxicity, such as anaemia, leukopenia or thrombocytopenia [5,22]. Grade 3–4 haematological toxicities were not observed in an open-label prospective phase II trial using ^177^Lu-DOTATATE [30], but in another study, these toxicities occurred in 32.1% of patients, although none developed myelodysplasia or haematological neoplasms [31]. In a different series including 1631 PRRT-treat patients, 1.8% had therapy-related myeloid neoplasms after a median time of 43 months [32]. Other adverse effects of PRRT include nausea (mainly caused by the amino-acid infusion), vomiting, mild abdominal pain and temporary hair loss [22,33].

## 3. PRRT for Aggressive or Metastatic Pituitary Neuroendocrine Tumours (PitNETs)

### 3.1. Aggressive and Metastatic PitNETs

Aggressive PitNETs are tumours that display an unusually rapid growth rate or clinically relevant growth rate despite the optimal use of conventional therapies [34]. However, an agreed definition for aggressive PitNET is currently still lacking [1,2]. The prevalence of aggressive PitNETs varies across studies, but overall, such cases remain relatively rare with less than 2% of pituitary macroadenomas showing an aggressive course [35]. In rare circumstances, PitNETs develop either craniospinal or systemic metastasis (more commonly to the liver, followed by cervical lymph nodes, bones and lungs), being commonly known as pituitary carcinomas, or more recently as metastatic PitNETs. Metastatic PitNETs represent <0.2% of all pituitary tumours, mainly derive from corticotroph or lactotroph tumours, and are associated with a poor prognosis [1,2,34].

Many studies have sought to identify markers of aggressive PitNET behaviour; however, only few aggressiveness markers are recognised, including radiological invasion (into the cavernous/sphenoid sinus or bone erosion), the proliferation markers Ki-67, mitotic count and p53 staining, as well as some histotypes such as sparsely granulated somatotroph tumours, null-cell or Crooke’s tumours [1,36,37,38,39]. To improve the prognostic assessment of PitNETs, a clinicopathological classification was proposed [37], and validated in further studies [37,40,41,42,43,44]. Other markers of aggressive PitNET disease have been investigated, including in the following domains: (i) genetics, with germline mutations in *AIP* and *MEN1* genes, or duplications in *GPR101*, as well as somatic mutations in genes *ATRX* or *TP53*, leading to more invasive and resistant PitNETs [45,46,47,48,49,50]; (ii) transcriptional and post-transcriptional regulation [51,52,53,54]; (iii) serum inflammation-based biomarkers [55,56,57]; and (iv) tumour microenvironment [58,59,60]. Nevertheless, no single marker is able to predict aggressive or recurrent PitNETs, and most of the above-mentioned markers still need validation in predicting aggressive or malignant PitNET disease.

The therapeutic goals in patients with aggressive or metastatic PitNETs are to control tumour growth and mass effects, as well as to control pituitary hormone excess syndromes, while at same time, preserving as much as possible, the patient’s quality of life and prolonging the progression-free survival (PFS) and overall survival. The management of aggressive or metastatic PitNETs is challenging as, by definition, such cases do not optimally respond to conventional treatments such as surgery, standard medical therapy or radiotherapy [2,34]. Further options beyond conventional treatments are limited, and at present, mainly include temozolomide [2,34,61]. Other unconventional medications currently being investigated and/or experimentally used are anti-angiogenic drugs such as bevacizumab, mTOR inhibitors (everolimus), tyrosine kinase inhibitors (lapatinib and sunitinib), and more recently, immune checkpoint inhibitors such as ipilimumab and nivolumab [58,61,62,63,64,65,66,67,68,69]. Due to PRRT’s effectiveness in the management of NETs and other SSTR-expressing tumours, as well as to the high expression of SSTRs in PitNETs, PRRT has emerged as a possible option to treat patients with aggressive or metastatic PitNETs. 

### 3.2. Somatostatin, Somatostatin Receptors (SSTRs) and Somatostatin Analogues for PitNETs

The discovery of somatostatin as an inhibitory polypeptide hormone was soon followed by its therapeutic use in the form of somatostatin analogues, which are now widely used to treat gastroenteropancreatic NETs, as well as patients with certain subtypes of PitNETs, particularly those hypersecreting growth hormone (GH), adrenocorticotropic hormone (ACTH) or thyroid-stimulating hormone (TSH) [70,71]. Somatostatin is an ubiquitous neuropeptide produced by the hypothalamus, which has inhibitory effects on the secretion of several hormones [70,71], as well as inhibits many biological functions like exocrine secretion, cell proliferation and angiogenesis [72,73]. The action of somatostatin is mediated by SSTRs [74,75,76,77]. There are five different SSTR subtypes (SSTR1 to SSTR5) widely expressed in many different tissues and organs, all of them bind to somatostatin with high affinity [70,71,72,77]. 

The anti-proliferative and anti-secretory effects of somatostatin raised great interest in the oncology field, particularly for NETs. The usefulness of somatostatin itself is limited due to its short half-life (~1.5 min), which has led to the development of somatostatin analogues with a higher stability and longer half-life. Despite their lower affinity for SSTRs in comparison to somatostatin, such analogues are effective in treating NETs [78], and also PitNETs [71]. Currently, there are three somatostatin analogues approved for clinical practice: octreotide, lanreotide and pasireotide [71,77,79,80].

The normal pituitary, as well as PitNET tissues, express in general all five SSTRs (Table 1). The expression of the different SSTR subtypes varies according to the subtype of PitNET [70,81,82,83,84,85,86,87,88]. GH- and TSH-secreting PitNETs mainly express SSTR2 (with more than 90% expressing SSTR2), but they also express SSTR5 to a lesser extent. The predominant subtype in ACTH-secreting PitNETs is SSTR5, but they also express SSTR2. Gonadotroph tumours mainly express SSTR3, and in prolactinomas, SSTR1 and SSTR5 are the most representative subtypes (Table 1) [70,89,90]. Expression levels of SSTRs, namely SSTR2, correlate with the response to somatostatin analogues [86,87,88,91,92], while resistance to these drugs was predominantly found in PitNETs lacking SSTR2 [91,92].

Effective targeting of SSTRs with high-affinity somatostatin analogues allowed the emergence of a theranostic approach centred on radiolabelled somatostatin analogues using radioisotopes with diagnostic purpose (^111^In and ^68^Ga), but also with therapeutic intention (^90^Y and ^177^Lu) [5,6]. The expression of SSTRs in PitNETs can be effectively evaluated using functional SSTR imaging, which helps predicting the applicability of PRRT [5,6]. Positive uptake on ^111^In-octreotide-scintigraphy has been reported in more than two-thirds of patients with GH- and TSH-secreting PitNETs, in at least half of the patients with non-functioning PitNETs (NF-PitNETs), and in about 40% of prolactinoma patients [93]. ^68^Ga-DOTATOC and ^68^Ga-DOTATATE positron emission tomography imaging represent a major advance when compared to ^111^In-octreotide-scintigraphy [6,93], with the quantitative radioisotope uptake correlating positively with SSTR2 expression in the respective tissues [94]. 

### 3.3. Data on Clinical Use of PRRT for Aggressive or Metastatic PitNETs

Despite the wide evidence of SSTRs expression in PitNETs, PRRT has been rarely used in the management of patients with aggressive or metastatic PitNETs. To the best of our knowledge, so far, the employment of PRRT has been reported only in 30 published cases [61,95,96,97,98,99,100,101,102,103,104,105,106,107,108], summarised in Table 2. From the 30 reported patients treated with PRRT, 23 (82%) had aggressive PitNETs and 5 (18%) had carcinomas (malignancy status was not provided in 2 cases [103]): 11 had NF-PitNETs [61,96,97,99,101,102,104,105]; 5, lactotroph tumours [61,95,96,101]; 3, somatotroph tumours [101,106,107]; 2, corticotroph tumours [98,108]; 1, somatolactotroph tumour [102]; 1, thyrotroph tumour [61,104]; and 9, unknown subtype (Figure 2). 

Of the 27 published cases with available information regarding the response to PRRT, 5 (18%) showed a partial response [61,95,104,106], 8 (30%) demonstrated stable disease, and 14 patients (52%) had progressive disease (Figure 2 and Table 2). Notably, some responses to PRRT are remarkable and long-lasting, as illustrated in a female patient with an aggressive prolactinoma that shrunk markedly following five cycles of PRRT, reducing from 63.1 mL (volume pre-PRRT) to 28.2 mL (after two cycles) and then to 15.3 mL (after four cycles) [95]; the long-term efficacy of PRRT was proven in this case, with a further noteworthy reduction down to 3.1 mL over the following 7 years, with a PFS after PRRT estimated at 108 months [96]. In this case, the initial tumour was resistant to cabergoline (0.5 mg/daily); however, cabergoline was resumed in the same dosage after PRRT [95,96]. Thus, it is unclear as to whether the observed long-term efficacy is only attributable to PRRT, or there may be a synergistic effect between PRRT and cabergoline. Interestingly, a synergic effect between PRRT and checkpoint inhibitors was documented in a patient with a multi-treated corticotroph carcinoma [69,108]. This case initially responded well to ipilimumab and nivolumab [69], but eventually escaped and the disease progressed, requiring a new therapeutic approach that consisted of four cycles of PRRT [108]. Immediately following the fourth cycle, her disease stabilised and nivolumab was resumed; remarkably, a 60% reduction of the tumour mass was seen 6 months later, accompanied by a decrease in serum ACTH levels [108]. This case suggests that checkpoint inhibitors (which have been increasingly used in pituitary tumours [58,62,63]) and PRRT-derived radiation may be complementary for the management of aggressive or metastatic PitNETs. It is possible that PRRT radiation-related cell lysis uncovers antigenic sites, triggers the release of proinflammatory cytokines, and/or has immunomodulatory effects, resulting in an immunogenic phenotype and sensitising PitNETs to checkpoint inhibitors, thereby augmenting the effectiveness of immunotherapy in PitNET patients, as already seen for other cancers [109,110].

More than 50% of the published cases had progressive disease after PRRT (i.e., did not respond to PRRT), despite these pituitary tumours had, in general, a significant expression of SSTRs on functional imaging prior to PRRT [93,96,111], suggesting that other *properties* than tumour surface SSTR2 expression and binding are needed for the tumouricidal effects of PRRT. SSTR functional imaging before PRRT is still needed to identify the expression of SSTRs in aggressive PitNETs, thus confirming the suitability for the treatment; however, challenges remain in predicting its efficacy as functional imaging is not able to adequately predict the response to PRRT. Another observation that can be made from the published non-responder cases is that PRRT seems to show no benefit for rapidly progressive disease, which often undergo few PRRT cycles (one or two) that may be insufficient to stabilise the disease or induce a response; consistently, a low Ki-67 proliferation index has been associated with better PRRT outcomes [112]. Maclean et al. reported a patient with an aggressive GH/PRL-secreting PitNET who had significant progressive disease and died prior to the third PRRT cycle, and another case of a silent corticotroph tumour ‘unsuccessfully’ treated with PRRT for rapidly progressive disease, although only one cycle was given [102]. Six more cases labelled as progressive after only 1 or 2 PRRT cycles were reported [61,96,98,100,101]. Therefore, in total, 8 of 14 cases with progressive disease after PRRT (57%) were actually submitted to only 1 or 2 cycles of PRRT, which may reflect a fruitless selection for PRRT in highly aggressive and rapidly progressive tumours that possibly would progress regardless of the type of treatment. Ultimately, this may lead to a critical underestimation of the efficacy and usefulness of PRRT in the management of aggressive (but not rapidly progressive) pituitary tumours. Conversely, patients with good performance status and slowly progressive disease may have more potential to benefit from PRRT, including stabilising tumour growth if shrinkage is not achieved [102]. 

The response or resistance to PRRT appears not to be related with patient’s age or gender, neither with the radionuclide/peptide or dosage used [96,111], nor with the PitNET subtype [61]. In fact, partial responses were observed in functioning and non-functioning tumours [61,95,96,106]. However, Giuffrida et al. described an association between PRRT resistance and previous treatment with temozolomide [96]. Currently, 3 out of 5 published patients with partial response after PRRT (60%), and 6 out of 8 patients with stable disease after PRRT (75%) were actually naïve for temozolomide, whereas among the 14 cases with progressive disease, at least 11 (79%) had previous temozolomide. These figures align with the Giuffrida et al. findings correlating a previous exposure to temozolomide with less responsiveness to PRRT. Although it is unclear if there is a biological mechanism explaining this association, or if instead, there is some sort of patient selection bias as tumours previously unresponsive to temozolomide might be more aggressive and less sensitive to other forms of treatment, including PRRT. Theoretically, PitNETs that progressed despite recent radiotherapy may be relatively radio-resistant, and thus less likely of benefiting from PRRT; additionally, changes in the blood supply following radiotherapy may also limit the delivery of PRRT to the pituitary tumour [102]. Nevertheless, prior radiotherapy should not preclude the use of PRRT, as there are cases previously submitted to external radiotherapy who had a partial response or stable disease after PRRT [61,95,97,102,106].

In general, PRRT has been well tolerated, and no major side effects are reported in patients with aggressive or metastatic PitNETs submitted to PRRT [61,104], apart from transient cytopenia [102,106], and an increase in facial pain following the first PRRT cycle in one case, although imaging studies pointed the PitNET progression as its likely cause [102]. There is also a report of a patient who suffered pituitary apoplexy about 1 year after PRRT [107], raising the question as to whether apoplexy could be a late consequence of the radiation exposure during PRRT. However, no other reports of pituitary apoplexy following PRRT have been described, namely in PitNET patients with follow-up longer than 3 years after PRRT [95,97,102,105]. Komor et al. reported intact pituitary function in a NF-PitNET patient more than 8 years after PRRT [97], while Maclean et al. described a patient with a non-functioning carcinoma who had PRRT remaining with stable disease for 40 months and required no adjustment to his replacement therapy [102]; also, the residual pituitary function has not changed over the course of 9 years since the administration of PRRT to an aggressive prolactinoma [95,96]. These reports, together with the data from studies on NET patients (discussed below) [33,113,114,115], suggest that the residual pituitary function may not be significantly affected in patients with aggressive or metastatic PitNETs after PRRT.

According to the latest guidelines on aggressive or metastatic PitNETs, PRRT may be considered as a therapeutic option for a patient with an aggressive tumour in case other treatments are not feasible or have failed in controlling the disease progression [34]. Overall, 30% of the published cases achieved stable disease while 18% showed partial response (Figure 2); hence, the likelihood of a positive outcome following PRRT (i.e., stable disease or partial response) for a patient with an aggressive, refractory and progressive pituitary tumour may be as high as near 50%, with up to one-fifth of cases potentially showing tumour reduction. Such a response rate is not very different from the other (few) therapeutic options for patients with aggressive PitNETs, including temozolomide and other systemic therapies [34,61,96,103]. Thus, PRRT can be regarded a safe option to consider in a patient with an aggressive or metastatic PitNET after demonstrating high expression of SSTRs through pre-PRRT functional imaging. 

In order to further understand the role and usefulness of PRRT in this setting, as well as to optimise PRRT-related response rates and outcomes, clinical research and prospective studies are needed. Future research could potentially explore different avenues, including: (i) the role of the earlier use of PRRT in the management algorithm of patients with aggressive PitNETs; (ii) the use of new radioligands such as radiolabelled somatostatin antagonists or radioligands with affinity for other SSTRs rather than only SSTR2 (possibly an universal radioligand able to bind all SSTRs); (iii) the combined use of PRRT with other medications, such as drugs able to upregulate the expression of SSTRs or drugs that may have synergic effects such as immunotherapy; (iv) the potential interaction and effects of previous exposure to temozolomide and other treatments such as radiotherapy in determining the efficacy and safety of PRRT; (v) the assessment as to whether the extent of isotope uptake on pre-PRRT functional imaging correlates or predicts the therapeutic response.

## 4. Pituitary Function following PRRT

SSTR density in normal endocrine organs is not as high as in NETs; however, the existence of SSTRs in such organs expose them to some degree of radiation during PRRT [77,115]. The normal anterior pituitary express SSTRs (Table 1), mainly SSTR2, SSTR5 and SSTR1 [74,75,76]. The expression of different SSTR subtypes varies according to the pituitary cell type, with rat studies suggesting that the highest expression of SSTR2 can be found in somatotrophs and thyrotrophs [74,116]. Consistently, the anterior pituitary shows the uptake on SSTR imaging, suggesting that pituitary cells may be exposed to radiation during PRRT [6,93]. Hypopituitarism secondary to external beam radiotherapy is widely recognised, and the risk increases proportionally with higher doses and long periods following the irradiation, with GH and gonadotrophin axes being more radiosensitive, whereas ACTH and TSH axes are significantly more resistant [117,118]. Hence, patients who received PRRT may be theoretically at risk of developing hypopituitarism, which has prompted some groups to investigate the effects of PRRT on the pituitary function. Four studies, summarised in Table 3, have assessed the PRRT effects on pituitary function in patients with non-pituitary neuroendocrine neoplasms who received PRRT [33,113,114,115]. Overall, the data from these studies suggest that there is no significant increased risk of clinically relevant hypopituitarism in patients exposed to PRRT [33,113,114,115].

### 4.1. Gonadal Axis

In post-menopausal women, two studies showed a decrease in serum follicle-stimulating hormone (FSH) and luteinising hormone (LH) levels following PRRT [114,115], which may suggest a potential effect of PRRT on gonadotrophs, although hypogonadotropic hypogonadism in post-menopausal women is not clinically relevant. In contrast, a previous study showed unchanged levels of LH and FSH (as well as oestradiol and inhibin B) before and 24 months after PRRT [33], and another study with a longer follow-up showed no differences in secondary hypogonadism rates between PRRT-treated and control females, with no differences in FSH levels between these two subgroups; moreover, none of the PRRT-treated post-menopausal women had inappropriately low levels of FSH and LH [113]. 

In men, gonadotrophins rise shortly after PRRT, accompanied by a decrease of inhibin B; however, both returned to baseline 18–24 months after PRRT [33,115]. Total testosterone decreases during the follow-up after PRRT [33,115], coincident with a decrease in sex hormone binding globulin (SHBG), but the biochemically active non-SHBG-bound testosterone does not change [115]. Elston et al. found no differences in the rates of secondary hypogonadism between PRRT-treated vs. control males, and PRRT was not a predictor of male hypogonadism [113]. 

Taking together these data, gonadal function may be subject to subtle changes following PRRT, but in the long-term, clinically relevant secondary hypogonadism seems not to be an issue for PRRT-treated patients. Men undergoing PRRT may suffer from a transient and reversible impairment of spermatogenesis [33,115], similar to male patients with thyroid cancer who undergo radioiodine therapy [119,120]. Although sperm analyses have not been systematically performed in the studies, the observation of remarkable decrease in inhibin B with concomitant raise in FSH, both returning to baseline several months after PRRT [33,115], suggest a temporarily impaired spermatogenesis. In fact, inhibin B, produced by the testicular Sertoli cells, plays a crucial role in the spermatogenesis, and is also a major (negative) feedback regulator of FSH [121,122,123].

### 4.2. Somatotroph Axis

The somatotroph axis, recognised as the most radiosensitive and the first pituitary axis to reflect the radiation effects [117,118], was assessed in PRRT-treated patients in two studies [113,114]. Sundlöv et al. showed a decrease in serum insulin-like growth factor 1 (IGF-1) levels following PRRT, estimated at −15% and −30% at 19–24 months and >48 months of follow-up, respectively, which correlated with the number of PRRT cycles and the absorbed radiation doses. This was interpreted as pituitary-related GH deficiency, as albumin levels during the follow-up did not change, making unlikely that IGF-1 decrease would be a result of liver damage; moreover, most patients were already receiving a fixed dose of somatostatin analogues at baseline which remained stable, therefore not explaining either the decrease of IGF-1 during the follow-up [114]. On a different study, 1 of 34 patients (3%) developed GH deficiency 55 months after PRRT, and Elston et al. reported a trend for GH deficiency based on lower IGF-1 Z-scores in PRRT-treated vs. control patients [113]. Whilst the development of GH deficiency may eventually explain constitutional symptoms, such as fatigue or body composition changes that might be reported by PRRT-treated patients, a finding of GH deficiency post-PRRT would not alter the management, as GH replacement therapy is contraindicated in patients with active malignancy [124], and neither should it preclude clinicians prescribing PRRT for a progressive and potentially threating neuroendocrine neoplasm.

### 4.3. Thyroid Axis

Sundlöv et al. and Elston et al. reported no changes in the thyroid axis as a result of PRRT [113,114], while two earlier studies showed that free thyroxine (FT4) levels decrease 3 to 24 months after PRRT with no changes in TSH and T3 [33,115]. The development of primary hypothyroidism is also uncommon in PRRT-treated patients, occurring in only 3% of cases [115]. The levels of reverse triiodothyronine (rT3) also decrease after PRRT [115]. Considering the chronicity and severity of the underlying malignant disease and their impact on the thyroid axis, it is currently unknown as to whether such changes, particularly in serum FT4, are secondary to the effects of PRRT or instead due to a cancer-related non-thyroidal illness [115], neither it is clear if these patients at a longer term will evolve to hypothyroidism requiring thyroxine replacement therapy.

### 4.4. Hypothalamo–Pituitary–Adrenal Axis

The hypothalamo–pituitary–adrenal axis remained intact in patients submitted to PRRT across three studies investigating the adrenal function post-PRRT [113,114,115]. These studies mainly relied on measuring basal serum ACTH and cortisol, while Teunissen et al. studied the adrenal reserve with low dose ACTH stimulation tests. An adequate cortisol response (>550 nmol/L) on the ACTH stimulation test was seen in all patients before and 24 months after PRRT; however, the mean peak cortisol response before PRRT was higher than that after PRRT (909 ± 57 vs. 822 ± 35 nmol/L; *p* < 0.001). Whether this subtle difference in the mean peak stimulated cortisol after PRRT reflects any partial radiation-induced adrenal insufficiency or just a less stressful state of patients remains unclear; nevertheless, such a difference is not clinically relevant as any of the patients failed the ACTH stimulation test, and thus none of them would be diagnosed with adrenal insufficiency nor require glucocorticoid replacement therapy [115].

## 5. Conclusions

Aggressive pituitary tumours can be resistant to conventional treatments, including surgery, radiotherapy or medical therapy. Temozolomide is recommended as first-line therapy for aggressive or metastatic PitNETs, when conventional means failed; however, the rate of relapse after temozolomide in the long-term is high. Other therapeutic approaches are emerging as possible options for selected cases, including PRRT. The expression of SSTRs in PitNETs, together with the effectiveness of somatostatin analogues in some subtypes of PitNETs, paved the way for employing PRRT for aggressive or metastatic PitNETs. However, there are only a few cases where such treatment has been used; hence, data about efficacy and safety of PRRT in this setting are scarce. From the 30 published cases, it seems that PRRT may induce a partial response in up to a fifth of cases, stabilise the disease in about a third, while the majority of patients will not respond to PRRT and will have progressive disease (Figure 2). Regarding safety, PRRT seems to be safe for PitNET patients, and there is no increased risk for new-onset (or worsening) of clinically relevant hypopituitarism in patients with PitNETs or non-pituitary NETs who undergo PRRT. At present, PRRT can be considered as a therapeutic option for patients with aggressive or metastatic PitNETs if other approaches are not feasible or have failed in controlling the disease progression. However, to fully establish the role and usefulness of PRRT in the management of patients with aggressive pituitary tumours, further research and prospective studies, in particular, are needed.

## Figures and Tables

**Figure 1 cancers-15-02710-f001:**
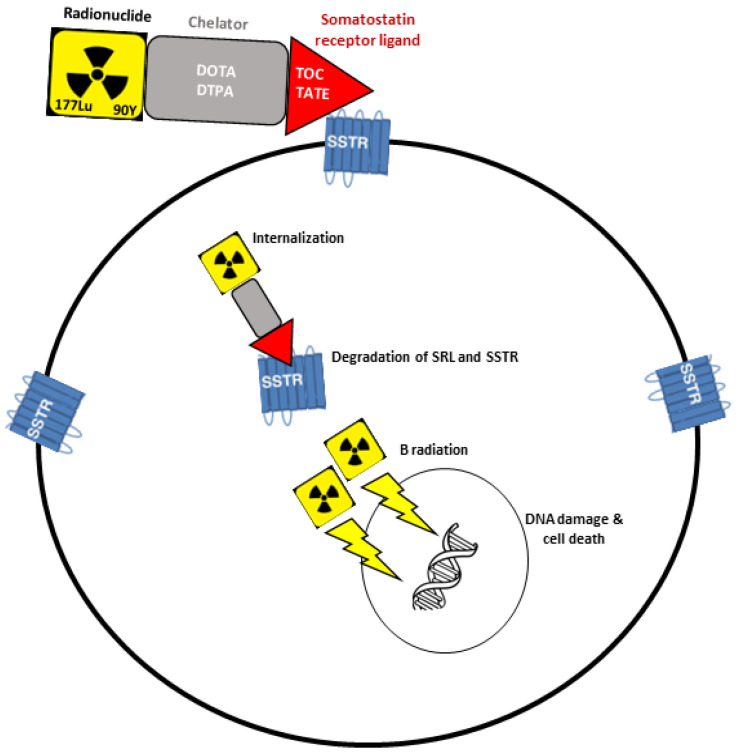
Principles of peptide receptor radionuclide therapy (PRRT). Therapeutic radiopharmaceuticals (^177^Lu or ^90^Y) are linked to the somatostatin receptor ligand (TOC or TATE) by a binding chelator (DOTA or DTPA), which stabilises the radioisotope and avoids its dissociation in vivo before targeting the SSTR-positive tumour tissues. The somatostatin receptor ligand will then bind to the SSTRs in the surface of the tumour cell, after which the complex is internalized. Once inside of the cell, degradation of the SSTR and the somatostatin receptor ligand will occur, and the radiation will be delivered to the double stranded DNA causing damage and cell death. DNA, deoxyribonucleic acid; DOTA, tetraazacyclododecane-tetraacetic acid; DTPA, diethylenetriamine pentaacetic acid; Lu, Lutetium; SRL, somatostatin receptor ligand; SSTR, somatostatin receptor; TATE, tyr3-octreotate; TOC, tyr3-octreotide; Y, Yttrium.

**Figure 2 cancers-15-02710-f002:**
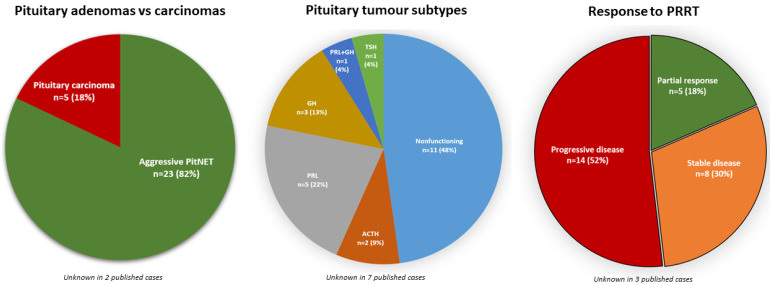
Aggressive or metastatic pituitary tumours treated with PRRT. Data were compiled and analysed considering three aspects: (i) pituitary adenomas vs. carcinomas; (ii) pituitary tumour subtype; and (iii) response to PRRT. Data are depicted in the graphs as the absolute number of patients and the respective percentages in brackets, concerning only the cases where that specific information were available; published case reports that omitted the respective data were not considered in the graphs, and were counted as “unknown”. The number of cases with “unknown” data are shown below each graph. ACTH, ACTH-secreting pituitary tumour; GH, GH-secreting pituitary tumour; PitNET, pituitary neuroendocrine tumour; PRL, PRL-secreting pituitary tumour; PRL + GH, PRL and GH co-secreting pituitary tumour; PRRT, peptide receptor radionuclide therapy; TSH, TSH-secreting pituitary tumour.

**Table 1 cancers-15-02710-t001:** Expression pattern of somatostatin receptors in the normal and neoplastic pituitary.

	SSTR1	SSTR2	SSTR3	SSTR4	SSTR5
Normal pituitary gland					
Foetal pituitary	+	+	+	+	+
Adult pituitary	+	+	+	-	+
Pituitary tumour					
GH-secreting PitNET	60%	90%	45%	<5%	90%
ACTH-secreting PitNET	60%	75%	10%	30%	75%
Prolactin-secreting PitNET	90%	60%	20%	0%	80%
TSH-secreting PitNET	100%	100%	0%	0%	50%
Non-functioning PitNET	25%	55%	45%	0%	50%

Adapted from Cuevas-Ramos D and Fleseriu M 2014 *J Mol Endocrinol* [70]. For the normal pituitary gland, data are shown as positive expression (+) or negative expression (−) of somatostatin receptors (SSTRs). For pituitary neuroendocrine tumours (PitNETs), data are shown as values representing the percentage of PitNETs per tumour type expressing each SSTR subtype. ACTH, adrenocorticotropic hormone; GH, growth hormone; PitNET, pituitary neuroendocrine tumour; SSTR, somatostatin receptor; TSH, thyroid-stimulating hormone.

**Table 2 cancers-15-02710-t002:** Summary of the published patients with aggressive or metastatic pituitary tumours treated with PRRT.

Reference (PMID)	Age/Sex	Tumour Type	Previous Treatments	Functional Imaging Prior to PRRT	Type of PRRT	Total Activity/Cycles Number	Response in Tumour Growth	Response in Hormone Reduction	PFS (Months)
Baldari 2012 *Pituitary* [95] (PMID: 22222543) # Giuffrida 2019 *Endocr Connect* [96] (PMID: 30939449) # Priola 2017 *World Neurosurg* [100] (PMID: 27713064) ¥	58/F	PRLoma	DA, Op, RT, SSA	^111^In-octreotide-scintigraphy	^111^In-DTPA-octreotide	37 GBq/5 cycles	PR-significant shrinkage over 9 yrs (from 63 to 3.1 mL)	Significant PRL decrease from 350,000 before PRRT vs. 30,310 U/L at last visit	108
Kumar Gupta 2012 *Int J Endocrinol Metab* [99] (PMID: 23843835)	71/F	NF-PitNET	None	^68^Ga-DOTA-NOC PET/CT	^177^Lu-DOTA-TATE	150 mCi/1 cycle	*na*	*na*	*na*
Kovács 2013 *Eur J Clin Invest* [98] (PMID: 23134557)	16/F	ACTH-PitCa	8xOp, BA, 3xRT	^111^In-octreotide-scintigraphy	^90^Y-DOTA-TATE	400 mCi/2 cycles	PD-died within the following year	No response	*na*
Komor 2014 *Pituitary* [97] (PMID: 23740146)	56/M	NF-PitNET	Op, RT	^111^In-octreotide-scintigraphy	^177^Lu-DOTA-TOC	600 mCi/3 cycles	SD > 8 yrs	not applicable	96
Maclean 2014 *Pituitary* [102] (PMID: 24323313)	63/M	NF-PitCa	2xOp, RT	^68^Ga-DOTA-TATE PET/CT	^177^Lu-DOTA-TATE	29.6 GBq/4 cycles	SD for 40 months, with CR in some metastatic nodules	not applicable	40
Maclean 2014 *Pituitary* [102] (PMID: 24323313)	42/M	GH/PRL-secreting PitNET	5xOp, RT, DA, SSA, TMZ	^68^Ga-DOTA-TATE PET/CT	^177^Lu-DOTA-TATE	15.3 GBq/2 cycles	PD-died shortly afterwards (prior to cycle 3 of PRRT)	*na*	0
Maclean 2014 *Pituitary* [102] (PMID: 24323313)	32/M	Silent ACTHoma	4xOp, RT, TMZ	^68^Ga-DOTA-TATE PET/CT	^177^Lu-DOTA-TATE	7.4 GBq?/1 cycle only due to facial pain	PD-died ~9 months later despite further Ctx, Op, TMZ and RT	not applicable	0
Bengtsson 2015 *J Clin Endocrinol Metab* [101] (PMID: 25646794)	59/F	NF-PitNET	TMZ	*na*	^177^Lu-DOTA-TATE	*na*	*na*	*na*	*na*
Bengtsson 2015 *J Clin Endocrinol Metab* [101] (PMID: 25646794) Burman 2022 *Eur J Endocrinol* [61] (PMID: 36018781) §	46/M	GH-PitCa	TMZ	Octreoscan	^90^Y-DOTA-TATE	Activity *na*/1 cycle	PD	*na*	*na*
Bengtsson 2015 *J Clin Endocrinol Metab* [101] (PMID: 25646794) Burman 2022 *Eur J Endocrinol* [61] (PMID: 36018781) §	23/M	PRLoma	RT, TMZ	^68^Ga-PET	^68^Ga-DOTA-TATE	Activity *na*/2 cycles	PD	*na*	*na*
Novruzov 2015 *Clin Nucl Med* [105] (PMID: 25275413)	68/M	NF-PitCa	Op, RT	^68^Ga-DOTA-TATE PET/CT	^177^Lu-DOTA-TATE	22.2 GBq/3 cycles	SD for 4 yrs	not applicable	48
Waligórska-Stachura 2016 *J Neurosurg* [106] (PMID: 26636388)	26/M	GH-secreting PitNET	Op, RT, SSA	^68^Ga-DOTA-TATE PET/CT	^90^Y-DOTA-TATE	400 mCi/4 cycles	PR-significant shrinkage at 12 months	IGF-1 decreased	12
Lasolle 2017 *Eur J Endocrinol* [103] (PMID: 28432119)	*na*	*na*	*na*	*na*	DOTA-NOC	*na*	PD	PD	*na*
Lasolle 2017 *Eur J Endocrinol* [103] (PMID: 28432119)	*na*	*na*	*na*	*na*	DOTA-NOC	*na*	Ongoing	Ongoing	*na*
McCormack 2018 *Eur J Endocrinol* [104] (PMID: 29330228) Burman 2022 *Eur J Endocrinol* [61] (PMID: 36018781) §	*na*	NF-PitNET	TMZ	Octreoscan	^90^Y-DOTA-TOC	Activity *na*/2 cycles	SD at 12 months	not applicable	12
McCormack 2018 *Eur J Endocrinol* [104] (PMID: 29330228) Burman 2022 *Eur J Endocrinol* [61] (PMID: 36018781) §	*na*	TSH-secreting PitNET	TMZ	Octreoscan	^177^Lu-DOTA-TATE	Activity *na*/1 cycle	PD	*na*	*na*
McCormack 2018 *Eur J Endocrinol* [104] (PMID: 29330228)	*na*	Aggressive PitNET (type not specified)	None	*na*	*na*	*na*	PR	*na*	*na*
McCormack 2018 *Eur J Endocrinol* [104] (PMID: 29330228)	*na*	Aggressive PitNET (type not specified)	None	*na*	*na*	*na*	SD	*na*	*na*
McCormack 2018 *Eur J Endocrinol* [104] (PMID: 29330228)	*na*	Aggressive PitNET (type not specified)	TMZ	*na*	*na*	*na*	PD	*na*	*na*
McCormack 2018 *Eur J Endocrinol* [104] (PMID: 29330228)	*na*	Aggressive PitNET (type not specified)	TMZ	*na*	*na*	*na*	PD	*na*	*na*
McCormack 2018 *Eur J Endocrinol* [104] (PMID: 29330228)	*na*	Aggressive PitNET (type not specified)	TMZ	*na*	*na*	*na*	PD	*na*	*na*
Giuffrida 2019 *Endocr Connect* [96] (PMID: 30939449) Priola 2017 *World Neurosurg* [100] (PMID: 27713064) ¥	54/M	PRLoma	DA, 3xOp, RT	^111^In-octreotide-scintigraphy	^177^Lu-DOTA-TOC	12.6 GBq/2 cycles	PD-increase after the 2nd cycle from 20 to 83.6 mL; then, TMZ and Ctx without benefit	*na*	0
Giuffrida 2019 *Endocr Connect* [96] (PMID: 30939449) Priola 2017 *World Neurosurg* [100] (PMID: 27713064) ¥	53/F	NF-PitNET	5xOp, RT, TMZ	^111^In-octreotide-scintigraphy	^177^Lu-DOTA-TOC	29.8 GBq/5 cycles	PD-increase from 7.7 to 14.1 mL	not applicable	0
Assadi 2020 *Eur J Nucl Med Mol Imaging* [107] (PMID: 31741022)	48/M	GH-secreting PitNET	Op, unspecific medical therapy	99 m-EDDA-HYNIC-tyr3-octreotide scintigraphy	^177^Lu-DOTA-TATE	22.2 GBq/3 cycles	SD over 1 year, then pituitary apoplexy	GH decreased; but IGF-1 remained high	12
Lin 2021 *J Endocr Soc* [108] (PMID: 34466766)	45/F	ACTH-PitCa	4xOp, 3xRT, SSA, DA, ketoconazole, CAPTEM, BA, Ctx, ICI			28.07 GBq/4 cycles	SD immediately after PRRT; nivolumab resumed after with shrinkage 6 months later (61% reduction)	ACTH decreased	6
Burman 2022 *Eur J Endocrinol* [61] (PMID: 36018781)	*na*	NF-PitNET	2xTMZ	^68^Ga-PET	^177^Lu-DOTA-TATE	Activity *na*/4 cycles	PR at 8 months	not applicable	8
Burman 2022 *Eur J Endocrinol* [61] (PMID: 36018781)	*na*	NF-PitNET	TMZ	Octreoscan	^177^Lu-DOTA-TATE	Activity *na*/4 cycles	PR > 26 months	not applicable	>26
Burman 2022 *Eur J Endocrinol* [61] (PMID: 36018781)	*na*	NF-PitNET	RT	Octreoscan	^177^Lu-DOTA-TOC	Activity *na*/6 cycles	SD	not applicable	*na*
Burman 2022 *Eur J Endocrinol* [61] (PMID: 36018781)	*na*	PRLoma	TMZ+ Bevacizumab	^68^Ga-PET	^177^Lu-DOTA-TATE	Activity *na*/1 cycle	PD	*na*	*na*
Burman 2022 *Eur J Endocrinol* [61] (PMID: 36018781)	*na*	PRLoma	2xTMZ	^68^Ga-PET	^90^Y-DOTA-TOC; ^177^Lu-DOTA-TATE	Activity *na*/2 cycles; 1 cycle	PD	*na*	*na*

# This patient has been reported on two different publications: in Baldari 2012 *Pituitary,* the short-term effectiveness of PRRT after 4 cycles was shown; in Giuffrida 2019 *Endocr Connect,* the long-term efficacy of PRRT during the following 7 years of this same patient was shown. ¥ Some data regarding these patients were briefly given in Priola 2017 *World Neurosurg*; however, the most detailed follow-up data concerning these were provided later in Giuffrida 2019 *Endocr Connect*. § These patients were first reported in other papers; however, further clinical and follow-up details were provided later in Burman 2022 *Eur J Endocrinol*. ACTHoma, corticotrophinoma; BA, bilateral adrenalectomy; CAPTEM, temozolomide + capecitabine; Ctx, chemotherapy; CR, complete response; CT, computed tomography; DA, dopamine agonists; GBq, gigabecquerel; GH, growth hormone; ICIs, immune checkpoint inhibitors; IGF-1, insulin-like growth factor 1; *na*, not available; NF-PitCa, non-functioning pituitary carcinoma; NF-PitNET, non-functioning pituitary neuroendocrine tumour; Op, operation; PD, progressive disease; PET, positron emission tomography; PFS, progression-free survival; PitCa, pituitary carcinoma; PitNET, pituitary neuroendocrine tumour; PMID, PubMed identifier; PR, partial response; PRL, prolactin; PRLoma, prolactinoma; PRRT, peptide receptor radionuclide therapy; RT, radiotherapy; SD, stable disease; SSA, somatostatin analogues; TMZ, temozolomide.

**Table 3 cancers-15-02710-t003:** Studies investigating the pituitary function following PRRT.

Reference (PMID)	Study Population	Gender/Mean Age	Previous Treatments	Type of PRRT	Activity/Number of Cycles	Follow-Up after PRRT	Main Findings Regarding the Pituitary Function Post-PRRT
Kwekkeboom 2005 *J Clin Oncol* [33] (PMID: 15837990)	131 pts with metastasized or inoperable gastroentero-pancreatic NETs	65 M, 66 F/56 years	48% had surgery; 5% EBRT; 15% chemotherapy; 50% SSA	^177^Lu-DOTATATE	600–800 mCi	24 months	- Serum TSH did not change during or after PRRT, while FT4 levels decreased significantly (mean 18.3 pmol/L before PRRT; and 15.5 to 17.5 pmol/L 3- to 24-months after PRRT) - In women: LH, FSH, estradiol and inhibin B levels did not change - In men: serum testosterone decreased in the follow-up period (from a mean of 14.4 nmol/L before PRRT to 10.4 nmol/L 24 months after the last cycle of PRRT; *p* < 0.01), while LH did not change. Serum inhibin B also decreased (from a mean of 179 ng/L before PRRT to 23 ng/L 3 months after the last cycle) accompanied by a rise in FSH, both returning to baseline at 18–24 months after the last PRRT cycle.
Teunissen 2009 *Eur J Nucl Med Mol Imaging* [115] (PMID: 19471926)	79 pts with various types of endocrine-related cancers (74 NETs, 4 thyroid cancers, 1 paraganglioma)	38 M, 41 F/54.8 years	46% had surgery; 8% chemotherapy; 4% EBRT; 46% SSA	^177^Lu-DOTATATE	600–800 mCi (3–4 cycles with 6- or 9-week intervals)	24 months	- 15 of 35 male pts (43%) had hypogonadism prior to PRRT - In men: serum inhibin B decreased 3 months after PRRT (205 ± 16 to 25 ± 4 ng/L; *p* < 0.05), suggesting transient spermatogenesis impairment; there was also an increase in FSH (5.9 ± 0.5 to 22.7 ± 1.4 IU/L; *p* < 0.05) and LH (5.2 ± 0.6 to 7.7 ± 0.7 IU/L; *p* < 0.05); these returned later near to the baseline level. Total testosterone and SHBG decreased (respectively, 15.0 ± 0.9 to 10.6 ± 1.0 nmol/L; *p* < 0.05, and 61.8 ± 8.7 to 33.2 ± 3.7 nmol/L; *p* < 0.05) while non-SHBG-bound testosterone did not change. - In post-menopausal women: serum FSH and LH decreased 24 months after PRRT (respectively, 74.4 ± 5.6 to 62.4 ± 7.7 IU/L; *p* < 0.05, and 21.1 ± 3.0 to 21.1 ± 3.0 IU/L; *p* < 0.05) - Thyroid axis: FT4 decreased 24 months after PRRT (17.7 ± 0.4 to 15.6 ± 0.6 pmol/L; *p* < 0.05), but TSH and T3 did not change (although tended to increase and decrease, respectively). rT3 decreased from 0.38 ± 0.03 to 0.30 ± 0.01 nmol/L (*p* < 0.05). Two of 66 pts (3%) developed primary hypothyroidism. - Adrenal axis: adequate cortisol response on ACTH stimulation tests in all pts before and after PRRT. Mean peak cortisol response before PRRT was higher than after PRRT (909 ± 57 vs. 822 ± 35 nmol/L; *p* < 0.001) - GH/IGF-1 axis was not evaluated
Sundlöv 2021 *Neuroendocrinology* [114] (PMID: 32259830)	68 pts with progressive grade 1–2 NETs	37 M, 31 F/66 years	80% had SSA; 12% chemotherapy; 15% biologics; 1% MIBG; 40% liver- therapies	^177^Lu-DOTATATE	Median 37.0 GBq (IQR: 14.8–66.6)	Median 30 months (range: 11–89)	- IGF-1 decreased during follow-up (*p* < 0.005): a decrease of −15% and −30% at 19–24 months and >48 months of follow-up, respectively. - Extent of IGF-1 decrease correlated with the number of cycles (*p* = 0.008) and with the absorbed radiation dose (*p* = 0.03) - In post-menopausal women, serum LH and FSH tended to decrease (*p* value NS) during follow-up, while in men, they increased in the first year following PRRT, after which returned to baseline - No changes in the adrenal or thyroid axes
Elston 2021 *Cancer Med* [113] (PMID: 34697905)	66 pts with unresectable metastatic NETs: 34 received PRRT vs. 32 controls	PRRT group: 23 M, 11 F/65.1 years vs. Controls: 15 M, 17 F/61.6 years	53% had SSA; 50% chemotherapy	*na*	Mean 31.8 GBq (IQR: 31.2–35)	Median 68 months (IQR: 51.3–102)	- There were no differences in male hypogonadism or other hormone deficiencies between PRRT-treated pts vs. controls - 16 of 38 pts (42%) men from the whole cohort had hypogonadism: 7 pts had primary hypogonadism (5 from the PRRT group); 9 pts had secondary hypogonadism (6 from the PRRT group) - No differences in the proportion of pts with secondary hypogonadism between PRRT-treated vs. control males (48 vs. 33%; *p* = 0.51), as well as among PRRT-treated vs. control females (0 vs. 12%; *p* = 0.51). PRRT did not predict male hypogonadism (OR = 1.8, 95% CI 0.5–7.1). - No differences in median FSH between post-menopausal women who had PRRT vs. those who did not (61.5 vs. 66 U/L) - No differences in the total dose received between PRRT-treated pts who developed secondary hypogonadism vs. those who did not (32.1 vs. 32.5 GBq; *p* = 1.000) - One of 34 PRRT-treated pts (3%) developed GH deficiency confirmed by both low IGF-1 and glucagon stimulation testing (55 months after PRRT, cumulative dose 33.7 GBq) - No differences in the proportion of pts with hyperprolactinaemia between PRRT-treated pts vs. controls (12 vs. 7%; *p* = 0.67) - No diabetes insipidus

ACTH, adrenocorticotropic hormone; CI, confidence interval; EBRT, external beam radiotherapy; F, females; FSH, follicle-stimulating hormone; FT4, free thyroxine; GBq, gigabecquerel; GH, growth hormone; IGF-1, insulin-like growth factor 1; IQR, interquartile range; LH, luteinising hormone; M, males; mCi, millicurie; MIBG, metaiodobenzylguanidine; *na*, not available; NET, neuroendocrine tumour; NS, not significant; OR, odds ratio; PMID, PubMed identifier; PRRT, peptide receptor radionuclide therapy; pts, patients; rT3, reverse triiodothyronine; SHBG, sex hormone binding globulin; SSA, somatostatin analogues; T3, triiodothyronine; TSH, thyroid-stimulating hormone.

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
