# Peer review of "The Effects of Peptide Receptor Radionuclide Therapy on the Neoplastic and Normal Pituitary"

_cancers, 2023, doi:10.3390/cancers15102710_

Round 1

Reviewer 1 Report

In this manuscript the authors reported a revision of published knowledge about PRRT in pituitary adenoma menagement.  I think that this paper could lead a substantial contribution to the knowledge of this potential therapy, not so considered in the present flow-chart of pituitary aggressive disease.

In my opinion, no particular revisions are necessary.

Author Response

I am grateful to the reviewer for the positive comment on this review manuscript.

(please find attached the full rebuttal letter)

Reviewer 2 Report

It is an interesting presentation of very important clinical topic of PRRT at the pituitary. My only concerns regard the length of the manuscript. Especiallly the section "principles...." is too long for me. Other editorial remarks: Table 1 after Table 2. table 2 difficult to read, too many data in the columns. Wrong location of the tables haedings (below the table). Wrong location of the Figure 2 (move to the left side). Please use dots ending the sentence after citations numbers, not before (line 189), please edit line 375. 

Author Response

I am grateful to the reviewer for the positive comment on this review manuscript.

Following the reviewer’s comment on the manuscript length, I have shortened the text, namely in the sections “Principles, usefulness and safety of peptide receptor radionuclide therapy (PRRT)” (as suggested by the reviewer), but also in the subsections “Aggressive PitNETs and pituitary carcinomas” and “Somatostatin, somatostatin receptors (SSTRs) and somatostatin analogues in PitNETs”.

Regarding the other editorial remarks of the reviewer: while the editorial/formatting process was not undertaken by me, including placing the tables and figures, I do acknowledge and agree with the remarks, and I am very grateful for bringing these up. Indeed, table 1 was placed after table 2, and far from the original text; table 2 has indeed quite a lot of data, but it is a key table as provides important details regarding each published case; Nevertheless, its current format has “squeezed” it so much that makes it difficult to read, hence I have formatted myself the Table 2 adapting it to the full page, which has reduced the number of pages and improve its readability. Worth noting that in order to address these editing issues, I have left comment boxes in the revised version of the manuscript specifically asking the editorial team of Cancers. The other minor aspects, included dots after the citation numbers, have been amended as well.

(please find attached the full rebuttal letter)

Reviewer 3 Report

Line 39: Rarely, PitNETs may also metastasize being then termed as pituitary carcinomas:

Please see updated classification:

Sylvia et al: Overview of the 2022 WHO classification of pituitary tumors.

  • PMID: 35291028
  •  
  • DOI: 10.1007/s12022-022-09703-7

Unlike the 2017 WHO classification, mammosomatotroph and acidophil stem cell tumors represent distinct PIT1-lineage PitNETs. The diagnostic category of PIT1-positive plurihormonal tumor that was introduced in the 2017 WHO classification is replaced by two clinicopathologically distinct PitNETs: the immature PIT1-lineage tumor (formerly known as silent subtype 3 tumor) and the mature plurihormonal PIT1-lineage tumor. Rare unusual plurihormonal tumors feature multi-lineage differentiation. The importance of recognizing multiple synchronous PitNETs is emphasized to avoid misclassification. The term "metastatic PitNET" is advocated to replace the previous terminology "pituitary carcinoma" in order to avoid confusion with neuroendocrine carcinoma (a poorly differentiated epithelial neuroendocrine neoplasm).

Line 44-46:peptide receptor radionuclide therapy (PRRT) specifically directed to neoplasms express- 44 ing high levels of somatostatin receptors (SSTRs), has proven effective and safe in patients 45 with NETs, and has been now emerging as a potential treatment option also for patients 46 with aggressive PitNETs or pituitary carcinomas

Suggest: Do not use "PitNETs or pituitary carcinomas" interchangeably. See highlighted text in the attachment.

Author Response

I am extremely grateful to the reviewer for providing this insight, supported by the article that he/she kindly shared. After reflecting on this, and in line with the recommendations of the reviewer and of the ‘new’ WHO 2022 classification, I have replaced the terminology “pituitary carcinoma” for “metastatic PitNET” whenever possible/appropriate, and I have explained this new usage in the text. However, occasionally I still use the term pituitary carcinoma, especially where referred by the original authors, as this is new terminology approach is still not widely known in the field and may lead to some unclarity or confusion by the future readers in some parts of this paper.

(please find attached the full rebuttal letter)
